# Music and mood regulation during the early stages of the COVID-19 pandemic

**Sarah Hennessy** [1☯‡], **Matthew Sachs**[2☯‡], **Jonas Kaplan**[1], **Assal Habibi**[1]*

**1** Brain and Creativity Institute, University of Southern California, Los Angeles, California, United States of America, **2** Center for Science and Society, Columbia University, New York, New York, United States of America

☯ These authors contributed equally to this work.
‡ Co-first authors.
* ahabibi@usc.edu

**Data Availability Statement:** All data files will be available upon publication from the Open Science Framework at DOI 10.17605/OSF.IO/HC65B, https://osf.io/hc65b/.

## Abstract

Music listening can be an effective strategy for regulating affect, leading to positive well-being. However, it is unclear how differences in disposition and personality can impact music's affective benefits in response to acute and major real-world stressful events. The COVID-19 pandemic provides a unique opportunity to study how music is used to cope with stress, loss, and unease across the world. During the first month of the spread of the COVID pandemic, we used an online survey to test if people from four different countries used music to manage their emotions during quarantine and if the functions of music depended on empathy, anxiety, depression, or country of residence. We found a positive relationship between the use of music listening for affect regulation and current well-being, particularly for participants from India. While people with stronger symptoms of depression and anxiety used music differently, the end result was still a positive change in affect. Our findings highlight the universality of music's affective potency and its ability to help people manage an unprecedented life stressor.

## Introduction

In March 2020, the COVID-19 outbreak forced people across the globe into quarantine. Communities around the world sought innovative ways to cope with growing anxiety, uncertainty, boredom, and social isolation. Music seemed to provide solace. Videos of apartment block performances, synchronous nightly cheering, and collective sing-a-longs from all over the world were shared widely across the internet. When Italian tenor Andrea Bocelli performed a solo Easter concert from an empty Milan Cathedral, the broadcast was viewed 35 million times.

These anecdotes suggest that across cultures, music listening helped people cope with the stress, uncertainty, and despair that stemmed from the pandemic. Music-listening can be beneficial for health and well-being across the lifespan [1–6], and listening to music reduces self-reported feelings of anxiety, and physiological measures of stress [7]. Music may also alleviate symptoms related to major psychological and mental disorders [8, 9] by helping people cope with negative affective states [10] and to balance their mood [11].

**Funding:** The Brain and Music Program (AH) at the Brain and Creativity Institute is supported by the GRoW at Annenberg Foundation (https://growannenberg.org/), the Los Angeles Philharmonic Association (https://www.laphil.com/) and the Van Otterloo Family Foundation. The funders had no role in study design, data collection and analysis, decision to publish, or preparation of the manuscript.

**Competing interests:** The authors have declared that no competing interests exist.

Another way in which music listening can impact well-being is through its ability to induce strong emotions [12–14]. Indeed, its emotional potency is consistently cited as one of the main reasons that people across demographic groups listen to music [15]. There are several mechanisms by which music-listening can alter one's affective states: by relaxing us [16, 17], strengthening emotional experiences [16], facilitating contemplation of emotional states [16], distracting us from a negative mood [18], or by allowing us to purge negative emotions [19]. Music can also alter affective states through reconnecting individuals to past memories [20, 21], providing feelings of social comfort [22], by connecting one's emotions to those of the performer [23], or through focusing on internal physical signals (for example, through entrainment to a musical beat) [24, 25]. Notably, music may also produce or enhance negative affective states [26, 27], and the degree to which individuals engage with music in a way that lowers well-being may be an indicator of proneness to mood disorders, such as depression [28].

Because of its emotional potency, music is often recognized as an effective tool for affect regulation, i.e., the process of changing the incidence, duration, and/or intensity of an affective state [29]. The Brief Music in Mood Regulation Scale (B-MMR) was created in 2012 to capture these various regulatory strategies [30], which includes seven subscales that cover multiple ways in which music may impact mood by both maintaining and enhancing positive moods as well as dealing with negative moods. Several of the subcomponents of the scale are associated with general emotion regulation strategies, such as reappraisal, and it has been shown that using music to reappraise leads to increased psychological well-being [31].

Interestingly, the Discharge subscale, which refers to selecting music that reflects one's current negative affective state in order to release or purge these feelings, is *not* positively correlated with reappraisal [32]. This subscale is also associated with symptoms of depression and anxiety [33]. However, it remains uncertain if individuals who are more anxious or depressed use music that reflects their affective state, for example, angry or sad music, to purge these emotions and feel better or if listening to this music leads to stronger feelings of anxiety or depression. The impact of music on mood likely depends on the goal of the user, the situation, individual differences, the type of emotion regulation strategy being used, and their pre-existing mood [16]. One individual difference measure that may influence this relationship is trait empathy. A particular component of empathy, Fantasy, which refers to the tendency to become absorbed into the situations of characters in stories, movies, and music, has been linked to the enjoyment of negative-valenced music [34]. It is therefore possible that the degree to which listening to sad or angry music can improve mood and well-being depends on this trait.

The COVID-19 pandemic provides a unique opportunity in which to study universal responses to a singular, unifying stressor on a grand scale. While laboratory studies have shown that music listening can aid in autonomic recovery after induction of acute stress [35–37] and trauma [38], there are relatively few studies on how individuals use music to regulate their emotions effectively in response to an acute major stressor, such as in the context of a significant, global event. And while there is evidence that certain cultural differences have been identified in the regulatory functions that music can serve, the evidence is clear that across the world, people choose to listen to music *because* of its affective function [39, 40]. Given these findings, we expect people to universally use music to regulate their emotions, but that the strategies, effectiveness, and qualities of the music-listening may differ across cultures.

It is worth noting that a number of research groups with similar interests have very recently published findings on the relationship between music listening and emotions during the early stages of the COVID-19 pandemic. An online questionnaire delivered to individuals living in 11 different countries found that music, in comparison to other forms of leisure activity, was

particularly effective at achieving certain well-being goals, such as venting negative emotions and maintain a good mood, across cultures [41]. This study also found that people who were feeling more distressed reported listening to negatively-valanced music, which was not found to be particularly effective in meeting their well-being goals, such as venting their negative emotions. Another recent study showed that individuals who are more likely to embrace a cognitive reappraisal strategy for regulating emotions reported listening to more happy and unfamiliar music during the pandemic, suggesting that certain types of music can help with reinterpreting and changing one's emotional response to a stressful situation [42]. Our study contributes to these findings by including a survey of the possible mechanisms by which music can influence emotion regulation and well-being (B-MMR) and including a more objective measure of music listening behaviors (acoustic analysis of songs submitted by participants).

Differences in acoustic features (for example, tempo, loudness, mode, texture) of music have been shown to predict perceived and felt emotions in listeners [43, 44]. The acoustic quality of the music that one chooses to listen has previously been shown to have implications for mental health status, where individuals with higher depression risk listen to music that is more repetitive and highly consistent in terms of "instrumentalness" [45]. Preferences for certain acoustic features (e.g., dynamics, tempo) have been additionally linked to personality factors (e.g., openness, conscientiousness) [46], and music-recommender systems that include personality factors (i.e., the Big Five) and acoustic features have higher accuracy than those that only include acoustic features [47]. Lastly, variations in acoustic features of chosen music can additionally highlight key differences in music listening habits across cultures [48, 49] and regional clusters of the world [50].

To this end, we conducted a large-scale, multi-national study with a culturally diverse population to explore the role of music listening in mood and emotion regulation during the early months of the COVID-19 pandemic. Through online surveys that asked participants to complete several well-validated questionnaires and to submit titles of music that they recently listening to, we aimed to investigate how people across the globe use music to regulate their mood and alleviate the discomfort and sadness of social isolation. There were four main hypotheses. 1) Across cultures, people who were more affected by the COVID-19 pandemic would be more likely to report using music to feel better. 2a) Emotion regulation strategies related to reappraisal and enhancing positive mood (both musical and general) would be associated with using music to feel better during the pandemic; 2b) Emotion regulation strategies related to dealing with negative emotions would be more effective in improving mood in individuals experiencing greater distress; 3) The relationship between music-listening emotion regulation strategies and improvements in mood and well-being would be moderated by trait differences in empathy. 4) These differences would be reflected in the acoustic quality of the music that people chose to listen to during the pandemic.

## Methods

This study and all protocols were approved by the University of Southern California Institutional Review Board (UP-20-00271). Data were collected and analyzed anonymously. The need to obtain informed consent was waived by the ethics committee; participants were instructed to exit out of the survey if they did not wish to participate.

### Participants

An online survey was distributed to individuals currently living in Italy, the United Kingdom, and the United States, through Prolific.co [51] on April 6th, 2020 and April 7th, 2020. Prolific.co is an online research platform with over 70,000 participants from around the world that are

**Table 1. Demographic and COVID-19 characteristics for study sample by country.** COVID-19 data reflects cases per 100,000 on the day the survey was distributed in each country, as reported by Johns Hopkins Coronavirus Resource Center.

| | | Total | India | Italy | United States | United Kingdom |
|---|---|---|---|---|---|---|
| *Gender* | | | | | | |
| | *n* | **589** | 155 | 136 | 148 | 150 |
| | *% Female* | **49** | 37 | 43 | 58 | 59 |
| *Age* | | | | | | |
| | *Mean* | **31.22** | 33.48 | 25.81 | 29.16 | 35.83 |
| | *SD* | **10.29** | 10.6 | 6.05 | 9.15 | 11.38 |
| *COVID-19* | | | | | | |
| | *Cases/100k* | | 2.2 | 231.0 | 130.6 | 92.2 |
| | *Deaths/100k* | | 0.01 | 28.56 | 5.11 | 11.34 |

vetted for their reliability. Prolific uses pre-screening filters to select participants for desired inclusion criteria such as language, age, or region. After removing participants for improper responses (i.e., missing the attention check or responding too quickly), this left 148 participants from the United States ($M_{age}$ = 29.16, 86 female), 136 from Italy ($M_{age}$ = 25.81, 59 female), and 150 from the UK ($M_{age}$ = 35.83, 89 female). Participants currently living in India (N = 155, $M_{age}$ = 33.48, 57 female) were recruited through using CloudResearch's Prime Panels between April 24th and April 27th. Prime Panels is an online research platform with more than 50 million participants across the globe who are screened based on response rate and accuracy level. On both platforms, additional inclusion criteria included being a citizen of the country in which one resides, being between the ages of 18–65, and being a proficient English speaker (self-reported fluency in English on Prolific and passed a standard English literacy and attention check on Prime Panels). This last requirement was necessary as not all of the questionnaires used here had been adapted to and validated in other languages. While a large portion of the populations of Italy and India speak English fluently, we acknowledge that our Indian and Italian samples are not representative of the entire populations of those countries but rather likely reflect a subpopulation of people who speak English and have higher levels of education than the average population. In total, 589 people completed the survey. For COVID-19 case rates at the time of data collection, see Table 1.

## Survey materials

The survey included several previously published questionnaires designed to assess a variety of individual difference measures. These questionnaires and a description of each are listed below:

1. Patient Health Questionnaire (PHQ) [52]: a brief measure of symptoms/severity of depression in the past two weeks.

2. State and Trait Anxiety Index (STAI) [53]: a state and trait anxiety measure

3. Emotion Regulation Quotient (ERQ) [54]: a widely used measure of two types of emotional regulation strategies, cognitive reappraisal, and expressive suppression.

4. Interpersonal Reactivity Index (IRI) [55]: a measure of empathy that assesses four subcomponents, including Fantasy, Perspective Taking, Empathic Concern, and Personal Distress.

5. Brief Music and Mood Regulation Questionnaire [32]: a 21-item self-report instrument for assessing the use of seven different music-related mood-regulation strategies. These include

a. Entertainment

b. Revival

c. Strong sensation

d. Diversion

e. Discharge

f. Mental work

g. Solace

In addition to these questionnaires, participants were asked about their music listening habits, both currently and a year ago (prior to the onset of the COVID-19 pandemic). Specifically, participants were asked how many hours they listened to music on an average day/a year ago and how many hours are they currently listening to music, what genre of music they most commonly listen to currently/listened to a year ago, and to list five songs that they are frequently listening to/listened to a year ago. While we understand that asking participants to retrospectively report their music listening habits from a year ago may be unreliable and subject to certain biases, we wanted to control for the fact that music preferences might change given the time of year, for example during the holidays. Therefore, in order to directly compare changes before and after the COVID-19 pandemic, we felt that it was best to ask about listening habits during the same time of year, a year prior. After listing songs that they are currently listening to, we directly asked participants the degree to which listening to music had made them feel better during the pandemic by asking how strongly they agree or disagree (1-strongly disagree to 5-strongly agree) with the following statements:

1. I have been turning to music listening or playing to take my mind off things

2. I find music listening or playing to be helpful in coping with the current crisis.

3. I have been listening to music more than usual to make myself feel better.

Because responses to these three questions showed high inter-item reliability (standardized Cronbach's alpha = 0.85), to calculate an overall individual score of the degree to which participants were using music to feel better, we took the mean response across all three questions. This dependent variable will subsequently to referred to as "use of music to feel better".

Participants additionally answered several YES/NO questions to assess distress, perceived risk, and change of mood related to the COVID-19 pandemic (S1A Text in S1 Text for specific questions). To assess the degree to which each individual was personally affected by the COVID-19 pandemic, an ordinal measure of COVID severity was calculated by totaling the number of YES responses for each participant to the four COVID-related questions in the survey (did you personally test positive, were you personally hospitalized, did you experience financial loss, are there shelter-in-place orders where you are) and adding this to the question regarding the degree of self-isolating (4–100%, 1—none). This measure of objective hardship related to COVID-19 will subsequently be referred to as COVID severity.

To assess changes in music listening habits and preferences during the COVID-19 pandemic, Spotify playlists were created that contained the songs that participants reported listening to pre-COVID and during COVID. Playlists were analyzed using a Python library to access data on Spotify's Web API https://github.com/plamere/spotipy. Auditory features were extracted from each song, which included:

1. *Acousticness*: real value between 0 and 1 measuring the likeliness that the track is acoustic

2. *Danceability*: real value between 0 and 1 measuring how danceable the song is

3. *Energy*: real value between 0 and 1 measuring intensity and activity;

4. *Valence*: real value between 0 and 1 measuring whether the song inspires positive/negative emotions

5. *Tempo*: real value expressed in beats per minute.

6. *Loudness*: real value between -60 and 0 measuring how loud the song is

7. *Mode*: integer (0 or 1) indicates the modality (major or minor) of a track, the type of scale from which its melodic content is derived (major is represented by 1 and minor is 0)

Finally, basic demographic information was collected, including a brief measure of socio-economic status (MacArthur Scale of Subjective Social Status; [56], years of musical training, age, and gender.

## Statistical analyses

All statistical analyses were performed using R [57]. A matrix displaying pairwise correlations between dependent measures and covariates (as well as between-task correlations) is available in S1 Fig. To assess the impact of COVID-19 on mood and music regulation, we used ANCOVA models (see Results for specifics). For all models, Age, SES, and education were included as covariates in the statistical models, given their correlations with many of the outcome measures. Gender was also included as a covariate given unequal distribution within and between countries. Musicianship was included as a covariate given that musician status may impact the emotional experience of music listening [58]. Lastly, country was included in all models. For the ANCOVA models, post-hoc comparisons between countries were conducted using Tukey's Honest Significant Difference. Given the number of statistical tests performed, p values were corrected using Benjamini & Hochberg's procedure [59] to reflect the false discovery rate (FDR). For all tests for significance, a corrected alpha level of 0.05 was used. For each model described, independent variables are denoted with (IV) and dependent variables are denoted with (DV).

## Results

While the objective of this study was to assess how people's music listening habits and well-being change due to the pandemic and its associated stress, regardless of their country of origin, we do not feel that the study is sufficiently powered to adequately address cross-country differences. That being said, for the interested reader, we do include statistical tests for differences between countries (IV) in COVID-19 perceived risk and severity (DV), as well as scores on the PHQ-9 (DV), STAI (DV), B-MMR subscales (DV) and ERQ (DV) in the S1B Text in S1 Text and S1 Table with the caveat that few conclusions can be drawn from such a small sample.

**H1**: *People more affected by the COVID-19 pandemic would use music to feel better.*

Per our first hypothesis, we assessed the degree to which distress and strain caused by the COVID-19 pandemic resulted in listening to music to improve mood (DV) using two separate 1-1-way ANCOVA models (perceived risk of COVID-19 as IV and COVID-19 severity as IV), controlling for age, gender, education, SES, and musicianship. Both measures of COVID risk were positively associated with reported improvements in mood by engaging music during the pandemic (personal COVID-19 risk: ($F_{(1, 544)} = 15.78$, $\beta = 0.16$, FDR-corrected $p < 0.001$, $\eta^2$

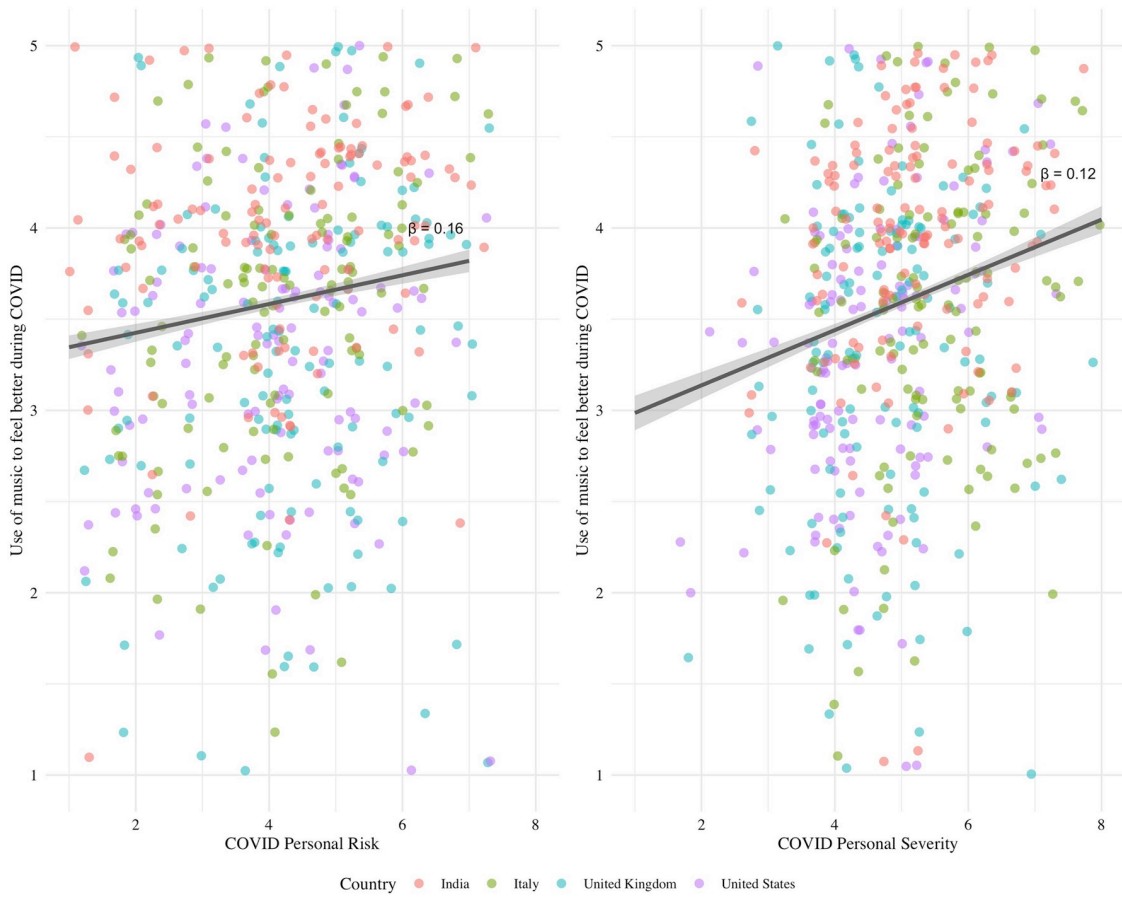

**Fig 1. Association between COVID personal perceived risk (left panel) and severity (right panel) and use of music to improve mood across countries.**

= 0.02), personal COVID-19 severity: (F(1, 544) = 7.24, β = 0.12 FDR-corrected p < 0.05, $\eta^2$ = 0.01), suggesting that the more personally affected by the pandemic, the more likely one turned to music to feel better (see Fig 1). We additionally tested the effects of COVID-19 personal risk in terms of the strategies in which music was used to regulate mood; these results are presented in S1B Text in S1 Text.

**H2a**: *Emotion regulation strategies related to reappraisal and enhancing positive mood (both musical and general) would be associated with using music to feel better during the pandemic.*

To test the second hypothesis, i.e., the degree to which scores on the B-MMR and ERQ predicted that music would make people feel better during the pandemic (DV), separate ANCOVA models with the same covariates were run for each subscale of the B-MMR (IV) and ERQ (IV). Across countries, total scores on the B-MMR significantly predicted that people would feel better after listening (F(1, 544) = 348.39, β = 0.62, p< 0.0001, $\eta^2$ = 0.03, see Fig 2). Interestingly, this positive relationship was found to be true for every subscale of the B-MMR as well, suggesting that all regulation strategies (both embracing positive emotions and dealing with negative emotions) were able to make people feel better.

Per our hypothesis, embracing a reappraisal strategy of emotion regulation (reappraisal subscale of the ERQ) also predicted using music to feel better (F(1, 544) = 30.27, FDR-

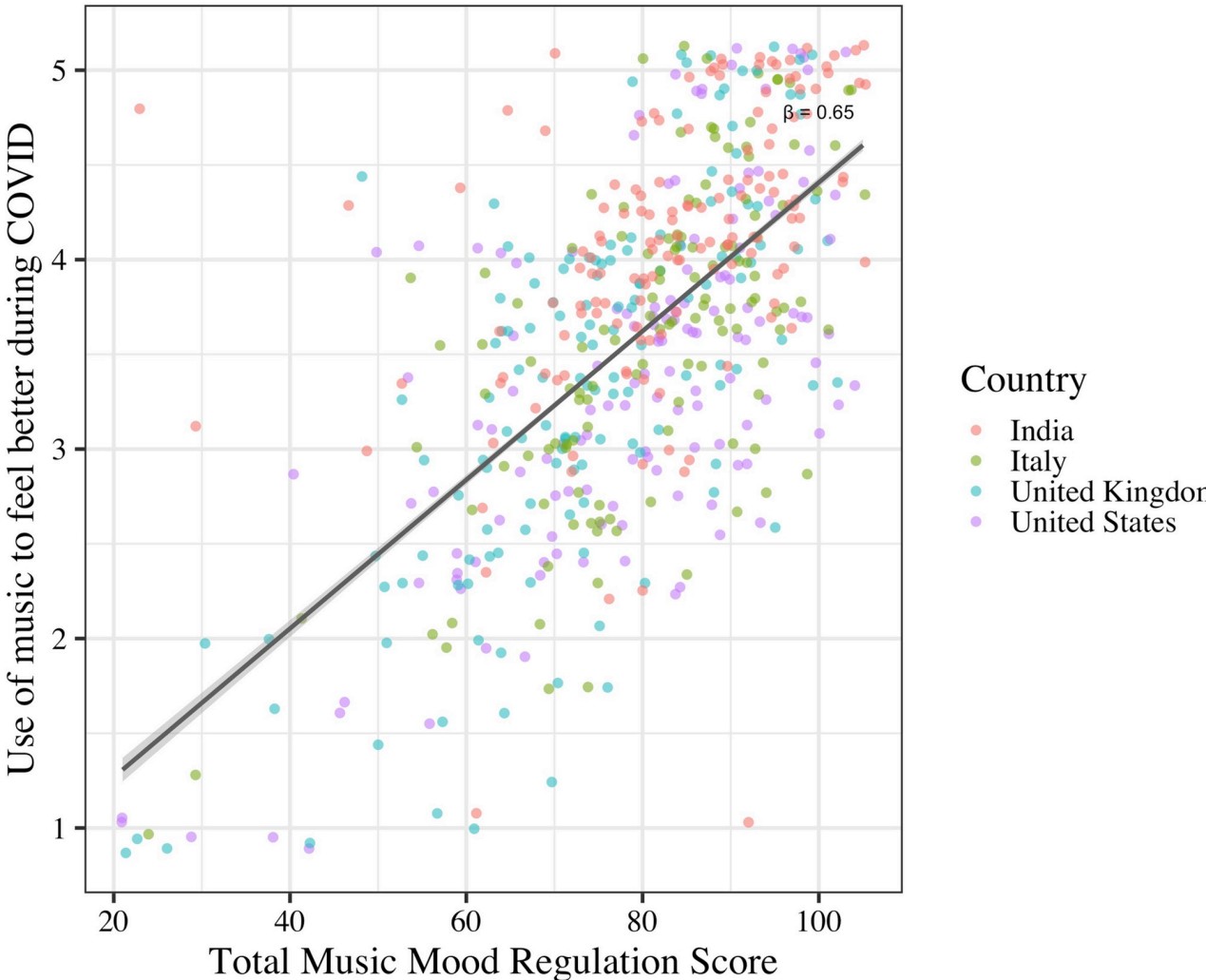

**Fig 2. Association of total music and mood regulation score (B-MMR total score) and use of music to feel better during COVID-19 across countries.**

corrected $p < 0.0001$, $\eta^2 = 0.05$, see Fig 3), where for all four countries, higher likelihood of reappraisal was associated with greater positive mood change ($\beta = 0.22$). The Suppression subscale did not predict use of music to feel better across ($F(1, 544) = 0.69$, $p > 0.05$, $\beta = 0.04$, $\eta^2 < 0.01$). Model results with ERQ subscales predicting each B-MMR subscale are presented in S1B Text of S1 Text.

***H2b***: *Emotion regulation strategies related to dealing with negative emotions would be more effective in improving mood in individuals experiencing greater distress.*

We conducted separate one-way ANCOVAs with the same covariates to assess whether depressive (IV) or anxiety (IV) symptoms predicted the use of the Discharge strategy (DV) and using music to feel better (DV). Depressive symptoms during COVID (PHQ-9 scores) predicted feeling better as a result of music listening ($F(1, 544) = 5.49$, $\beta = 0.10$, FDR-corrected $p < 0.05$, $\eta^2 = 0.00$), however this was not significant after FDR correction (FDR-corrected $p = 0.11$; see S1 Text for associations between PHQ-9 and additional regulatory strategies).

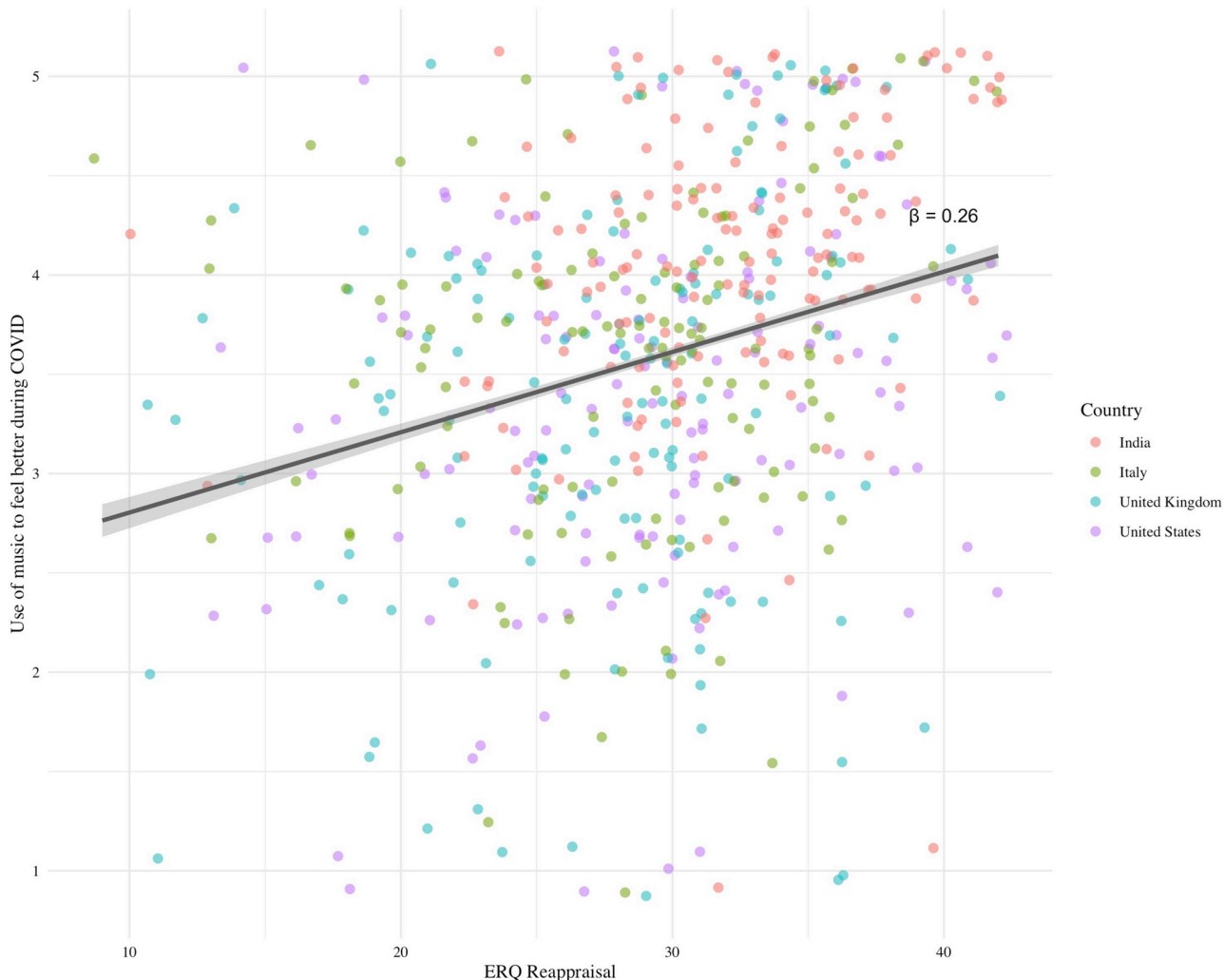

**Fig 3. Association between emotion regulation strategy (ERQ reappraisal) and use of music to feel better during COVID-19, across countries.**

The use of the Discharge strategy (see Fig 4) ($F_{(1, 544)}$ = 51.95, β = 0.30, FDR-corrected $p < 0.0001$, $\eta^2$ = 0.08) and overall B-MMR scores ($F_{(1, 544)}$ = 10.30, β = 0.14, FDR-corrected $p < 0.01$, $\eta^2$ = 0.02) additionally predicted feeling better as a result of music listening. However, against our hypothesis, no significant interaction was found between PHQ-9 scores and Discharge subscale in terms of using music to feel better ($F_{(1, 542)}$ = 0.66, β = -0.04, $p > 0.05$, $\eta^2 < 0.01$). We additionally tested whether changes in mood due to COVID-19 were predicted by depression symptoms, or the Discharge strategy of the B-MMR; these results are presented in S1B Text of S1 Text and S2 Fig.

State-level anxiety from the STAI did not predict the use of music to feel better ($F_{(1, 544)}$ = 2.29, β = 0.06, $p > 0.05$, $\eta^2 < 0.01$). Although we found that state anxiety positively predicted the use of the Discharge strategy ($F_{(1, 544)}$ = 17.70, β = 0.18, FDR-corrected $p < 0.001$, $\eta^2$ = 0.03 (see Fig 4), we did not find a significant interaction between feeling more anxious, using music to discharge negative emotions, and feeling better as a result of music listening ($F_{(1, 542)}$ = 1.07, β = 0.04, FDR-corrected $p > 0.05$, $\eta^2 < 0.01$). The trait component of the STAI is

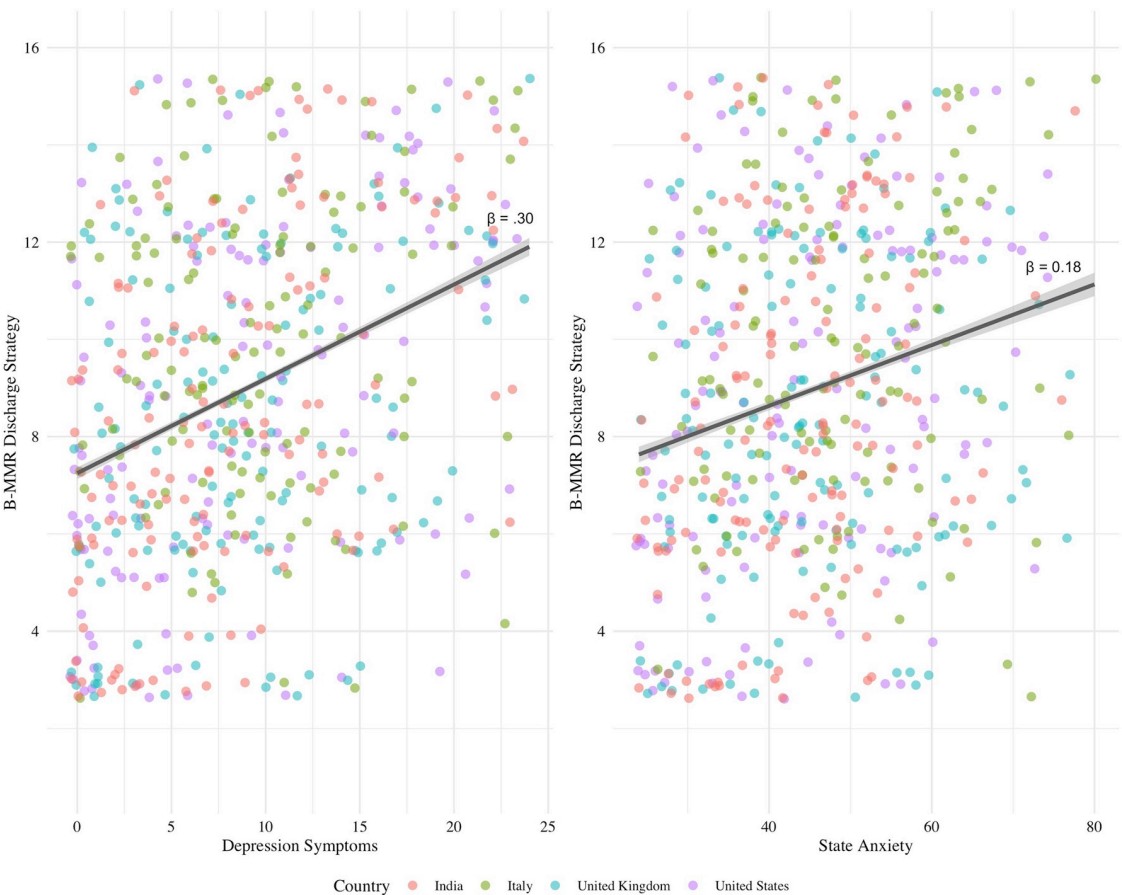

**Fig 4. Association between depression symptoms (PHQ score) (left panel) and STAI state anxiety (right panel) and B-MMR Discharge strategy across countries.**

less relevant in the context of the COVID-19 pandemic as a current stressor and is therefore presented in the S1B Text in S1 Text.

***H3***: *The relationship between music-listening emotion regulation strategies and improvements in mood and well-being is moderated by trait differences in empathy.*

We conducted separate one-way ANCOVAs using the same covariates to assess whether the Fantasy component of empathy (IV) predicted using music to feel better (DV) or B-MMR scores (DV). The Fantasy component of empathy was positively associated with using music to feel better during the pandemic ($F_{(1, 544)} = 22.94$, β = 0.19, FDR-corrected $p < 0.0001$, $\eta^2 = 0.04$). The Fantasy component of the IRI was also positively associated with total B-MMR scores ($F_{(1, 544)} = 34.00$, FDR-corrected $p < 0.0001$, β = 0.23, $\eta^2 = 0.05$), as well as all sub-scales. Given the connection between Fantasy-proneness and enjoying sad music, we tested if there was an interaction between Fantasy (IV) and Discharge (IV) on using music to feel better (DV), but this was not significant ($F_{(1, 542)} = 0.04$, β = -0.007, $p > 0.05$, $\eta^2 = 0.00$). To probe this further, we tested whether personal risk of COVID-19 might have influenced this previously found link. We found modest evidence for an interaction, that the relationship between Fantasy-proneness and using music to discharge negative feelings was weakened in people who felt more personally at risk of COVID-19 (see Fig 5). However, this was not significant

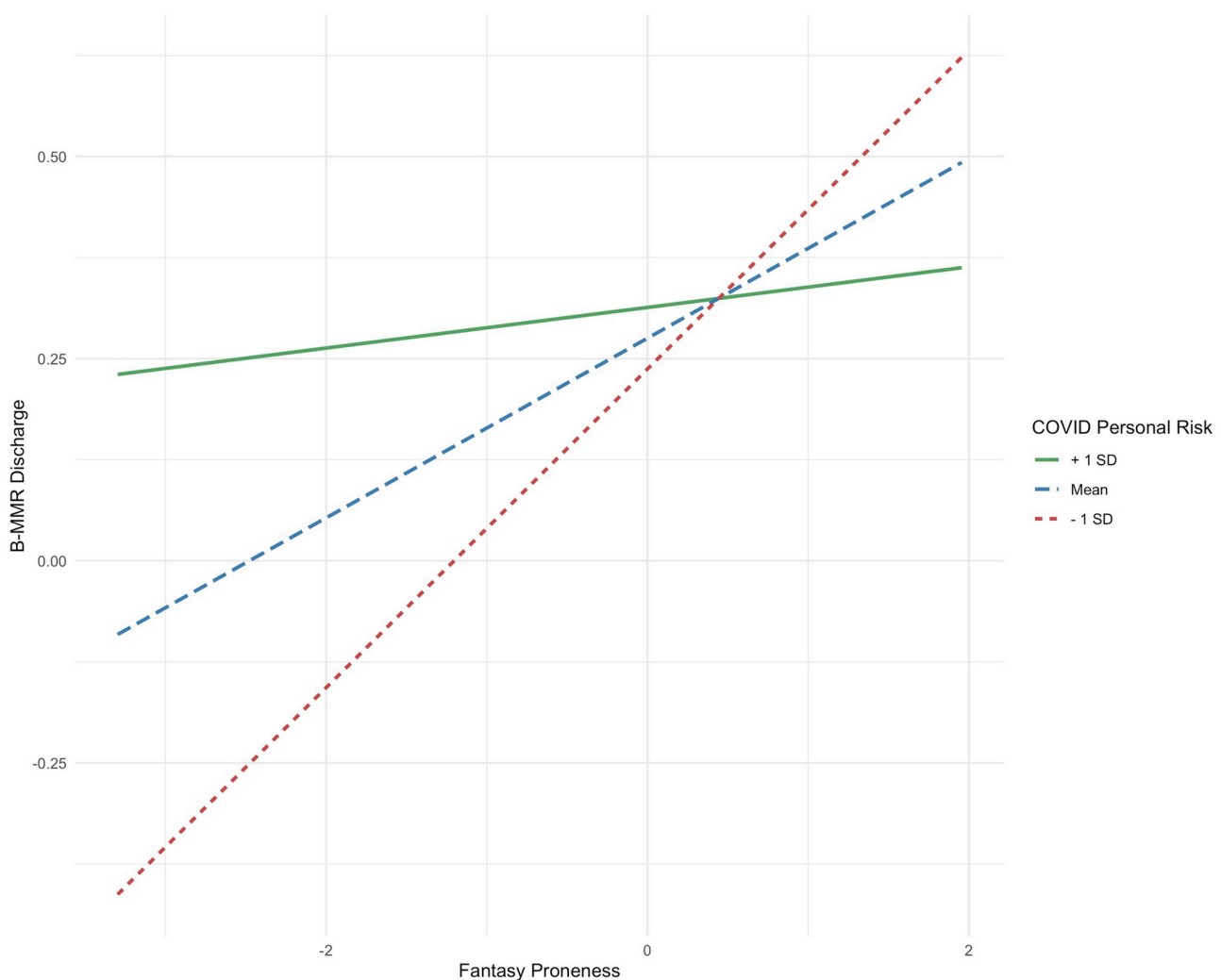

**Fig 5. Interaction between personal risk of COVID-19 and Fantasy-proneness to predict B-MMR Discharge strategy.**

after correcting for multiple comparisons (F(1, 542) = 4.63, β = -0.09, p < 0.05, FDR-corrected p = 0.15, $\eta^2$ = 0.01).

**H4**: *Changes in the acoustic quality of the music that people chose to listen to during the pandemic.*

To test our fourth hypothesis, separate repeated-measures ANOVAs were conducted on each acoustic variable (DV), with country (IV) as the between-subjects factor, and time (IV) (pre- or post-COVID) as the within-subjects factor. Additional analyses on the music selected pre and post COVID are included in the S2 Text.

No significant within-subject effects of time (pre versus post COVID) were found for any of the musical features extracted from Spotify's API. However, significant associations were found between the musical features of songs listened to during COVID and individual differences. Using the same covariates as above (age, education, gender, musicianship and country), we found that COVID severity was negatively associated with loudness (F(1, 416) = 4.99, β = -0.10, p = 0.02, $\eta^2$ = 0.01), energy (F(1, 416) = 6.33, β = -0.11, p = 0.01, $\eta^2$ = 0.01) and tempo (F

(1, 416) = 4.19, β = -0.09, p = 0.04, $\eta^2$ = 0.01), suggesting that people who are more strongly affected by COVID are choosing to listen to music that is calmer, quieter, and slower. However, these features, nor any other, did not predict a positive mood change as a result of music during the pandemic. No significant interactions were found between musical features, COVID-severity, and positive mood change. Additional playlist analyses, including changes in genre and exploration of songs falling within participants' "reminiscence bump," were also assessed (see S2 Text).

## Discussion

In this study, we attempted to quantify the ways in which music listening improved mood and well-being for people over the globe during the early months of the COVID-19 pandemic. We observed that, across four different countries, people more affected by the COVID-19 pandemic were more likely to report using music to feel better. Despite varying levels of COVID-19 severity and risk, we observed that using music to enhance positive moods and reappraise the situation predicted feeling better after listening to music, and people who were more affected by the pandemic or showed more symptoms of depression or anxiety were more likely to report using music to regulate their mood. Additionally, the relationship between certain negative music-related emotion regulation strategies, specifically Discharge, and improvements in well-being were moderated by trait differences in empathy. Lastly, these differences were reflected in the acoustic qualities of the music that people chose to listen to during the pandemic. These results suggest that music had a salubrious impact on people during a global crisis that transcended potential differences in culture and governmental response to the pandemic.

Across countries, participants who reported feeling more perceived risk of COVID-19 and those who were personally more impacted by COVID-19 showed greater improvements in mood after engaging with music during the pandemic. This suggests that people who are feeling particularly affected by the pandemic were more likely to turn to music to feel better. This supports recent findings that pleasant music led to decreased feelings of tiredness, sadness, fear, and worry in Italian healthcare workers in a COVID-19 hospital [60], and that music listening improved well-being and facilitated coping during the COVID-19 pandemic [41].

Multiple mood regulation strategies were related to positive well-being, though the use of these strategies varied with individual differences. Across all participants, emotion regulation strategies in both the musical and non-musical (i.e., cognitive reappraisal) domains, were associated with using music to feel better during the pandemic. We observed that all strategies of the B-MMR were associated with feeling better after music listening, suggesting that regulating both positive and negative emotions with music was effective at improving mood. Endorsement of reappraisal strategies, but not suppression, was also associated with positive mood change after listening to music, supporting previous findings that using music increases well-being when used to reappraised one's emotional situation [27, 61]. This additionally supports findings observed by Ferreri and colleagues [42], where individuals who were more likely to use cognitive reappraisal strategies sought out happy music to help regulate their emotions during the COVID-19 pandemic. Thus, individuals who regulate their emotions through re-interpretation of emotional situations, rather than through suppression of emotions, may be more likely to find that music is a particularly effective mechanism by which to feel better during times of unprecedented and global stress.

We additionally hypothesized that emotion regulation strategies related to dealing with negative emotions would be more effective at improving mood through music in individuals experiencing greater distress. Individuals who experienced actual hardships as a result of

COVID-19 (not perceived risk) reported using music to experience and regulate a range of emotions, such as finding solace, for mental contemplation, and feeling intense emotions, highlighting the different strategies we rely on when anticipating a stressful outcome versus experiencing that outcome. On the other hand, releasing negative emotions by listening to negative-valent music (Discharge) was positively associated with symptoms of state anxiety and depression. Given that our analyses are correlational, we cannot conclude whether people who are more anxious and depressed seek out music that conveys negative emotions for its therapeutic potential or that listening to negative-valent music actually increases feelings of anxiety and depression. While our results cannot directly resolve this uncertainty, there is evidence to suggest that the former interpretation is more likely. First, we found a positive relationship between depressive symptoms and the general use of music to improve mood during the pandemic (overall B-MMR scores). Second, we found a positive relationship between using music to discharge emotions in general and improved mood through music-listening during COVID specifically. And third, even though increased depression was associated with feeling worse during COVID, the relationship was weaker in people who regulate their mood with music through the discharge strategy. Combined, these results suggest that people who were stressed or sad during quarantine were able to use negative-valent music to feel better, or at least to feel less bad.

The Fantasy component of empathy was also positively associated with using music to regulate mood through the discharge of negative emotions as well as with using music to feel better during the pandemic. Our previous work showed that Fantasy was associated with the enjoyment of sad music *because* it was able to elicit strong, positive emotions, suggesting that Fantasy-prone individuals are particularly suited to benefit from a discharge strategy when listening to negative-valence music [34]. Interestingly, there is marginal evidence that feeling personally at-risk for contracting COVID-19 weakened the relationship between Fantasy and the discharge strategy, suggesting that the pandemic, and the stress associated with it, may attenuate the positive emotional benefits of mentally transporting into music. Whether listening to music congruent with one's negative mood is beneficial may therefore be contingent on the situation. Accordingly, in our previous study, Fantasy-proneness was positively associated with listening to sad music specifically when experiencing feelings of loneliness, but not when experiencing stress or anxiety [34]. It may be that Fantasy-prone individuals selectively use music to discharge when feeling general loneliness and sadness but find this strategy less useful during periods of intensified anxiety due to a personal COVID risk. We also note that our data were collected during the early stages of the pandemic, where knowledge about the virus was limited, and many people may have experienced uncertainty about how contracting COVID 19 might impact them. This uncertainty and inaccessibility to information to truly assess one's risk may have contributed to this relationship between Fantasy and the discharge strategy.

Lastly, this study assessed whether use of music to regulate mood would be reflected in the acoustic features of the music that people chose to listen to before and during the pandemic. While we did not find significant within-participant differences between various musical features of songs that people chose to listen to before and during the pandemic, we did find that people who were more severely impacted by the pandemic tended to listen to music that was less loud and energetic and more acoustic. Overall, the people who listened to softer, more acoustic music also reported feeling better as a result of listening to music during the pandemic. Interestingly, people who reported using music to discharge negative feelings actually preferred music that was *more* energetic and *less* acoustic, which is consistent with the goal of using music to purge or release negative emotions.

It is important to point out that the sample collected from each country may not be representative of the country's population. Specifically, the mean age of participants skewed low,

likely due to the online nature of the study. Additionally, the survey was administered in English to participants from countries that are primarily not English-speaking (i.e., Italy). Given that we did not include an English comprehension assessment, we cannot guarantee that all participants had the same understanding of survey questions. Lastly, even if English comprehension was identical across countries, this further reduces representativeness in countries whose primary language is not English. With these caveats in mind, we showed that across four different countries on three different continents, listening to music to regulate mood was a strong predictor of affective well-being during the COVID-19 pandemic. While the mechanisms by which music is able to improve mood may change across people, the fundamental result is the same. Music proves to be a powerful and salubrious tool during these unprecedented times.

## Supporting information

**S1 Fig. Pairwise pearson correlations for all behavioral measures of interest and covariates.**
(DOCX)

**S2 Fig. Interaction between PHQ-9 and B-MMR discharge on COVID-19 mood change.**
(DOCX)

**S1 Table. ANCOVA results, country predicting each Brief Music and mood regulation strategy.**
(DOCX)

**S1 Text. Supporting methods and results.**
(DOCX)

**S2 Text. Additional playlist analyses.**
(DOCX)

## Acknowledgments

We thank Amita Padiyar for her contribution to this project.

## Author Contributions

**Conceptualization:** Matthew Sachs, Jonas Kaplan, Assal Habibi.

**Data curation:** Sarah Hennessy, Matthew Sachs.

**Formal analysis:** Sarah Hennessy, Matthew Sachs.

**Writing – original draft:** Sarah Hennessy, Matthew Sachs, Assal Habibi.

**Writing – review & editing:** Jonas Kaplan, Assal Habibi.

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
