## [Decision Letter · Decision Letter 0]

5 Aug 2021

PONE-D-21-11175

Music and mood regulation during the early-stages of the COVID-19 pandemic

PLOS ONE

Dear Dr. Habibi,

Thank you for submitting your manuscript to PLOS ONE. After careful consideration, we feel that it has merit but does not fully meet PLOS ONE’s publication criteria as it currently stands. Therefore, we invite you to submit a revised version of the manuscript that addresses the points raised during the review process.

We look forward to receiving your revised manuscript.

Kind regards,

Stefan Koelsch

Academic Editor

PLOS ONE

Journal Requirements:

2. Thank you for including your ethics statement: "This study and all protocols were approved by the University of Southern California Institutional Review Board (UP-20-00271). Data were collected and analyzed anonymously. "

a) Please provide additional details regarding participant consent. In the ethics statement in the Methods and online submission information, please ensure that you have specified (1) whether consent was informed and (2) what type you obtained (for instance, written or verbal, and if verbal, how it was documented and witnessed). If your study included minors, state whether you obtained consent from parents or guardians. If the need for consent was waived by the ethics committee, please include this information.

“The Brain and Music Program at the Brain and Creativity Institute is supported by the GRoW at Annenberg Foundation, the Los Angeles Philharmonic Association and the Van Otterloo Family Foundation”

“The Brain and Music Program (AH) at the Brain and Creativity Institute is supported by the GRoW at Annenberg Foundation (https://growannenberg.org/), the Los Angeles Philharmonic Association (https://www.laphil.com/) and the Van Otterloo Family Foundation. The funders had no role in study design, data collection and analysis, decision to publish, or preparation of the manuscript.”

Reviewers' comments:

Reviewer's Responses to Questions

**Comments to the Author**

1. Is the manuscript technically sound, and do the data support the conclusions?

Reviewer #1: Partly

2. Has the statistical analysis been performed appropriately and rigorously? 

Reviewer #1: No

3. Have the authors made all data underlying the findings in their manuscript fully available?

Reviewer #1: Yes

4. Is the manuscript presented in an intelligible fashion and written in standard English?

Reviewer #1: Yes

5. Review Comments to the Author

Reviewer #1: The study considers an interesting topic and the writing style of the paper is good. However the paper needs considerable work to make it ready for publication. In particular I would recommend that the hypotheses to be tested be narrowed, particularly to focus on the findings relevant to music use in COVID, allowing the statistical analyses performed to also be more focused. The focus should be well justified in the introduction. At the moment the hypotheses as somewhat vague and the statistical tests are therefore numerous and rather exploratory in nature. More specific comments are below.p.3 – Reference needed to support the definition provided for “affect regulation”

p. 4 – Mechanisms by which music listening can alter affective states – this is by no means an exhaustive list and more literature could be cited here in order to provide a more comprehensive examination of the topic. E.g. plenty of research indicates that music can also alter affective states by reconnecting people with memories or via messages of hope in the lyrics, or can re-energise via processes of entrainment, it can also provide a sense of social connection and comfort. It is also worth noting the potential for music to cause negative alterations to affective states. Also the B-MMR was apparently developed in 2012, which does not seem too recent. A more recent scale was developed by the same author as the B-MMR, i.e. the HUMS (Healthy & Unhealthy Uses of Music Scale), which might be useful to note here too. The discussion about discharge and reappraisal warrants more detail as well. Refer to literature on coping styles and their relative effectiveness. Some quite deep topics are covered very superficially here and with insufficient reference to current research. It is unclear why the topic has suddenly shifted to that of listening to sad music.

A quick scan in Google Scholar has revealed numerous papers that have already been published on the topic of music and COVID-19, none of which have been discussed here. What does the current study add to the knowledge already shared in these studies?

p. 5 – the hypotheses are non-specific and vague. Is the first hypothesis a group comparison? For the second hypothesis, which traits would be associated with which strategies and why? This has not been well set up in the introduction. The tests addressing the second hypothesis are less interesting in the context of this research because they do not seem to be specifically related to COVID-19 and therefore the gap in the literature that is being addressed here needs to be more clearly explained in the Introduction.

Methods – More detail about the online platforms used to recruit participants is needed. Please clarify whether ‘native English speakers’ were specifically recruited or whether it was sufficient that they be proficient English speakers. In Italy, at least, this might have considerably narrowed the population to mostly people who were not born in Italy.

p. 10 – For the reader it is easier if the type of analysis performed is stated at the beginning of reporting the result of each test rather than listed under separate sub-headings in the Methods section. Also please state what the IVs and DVs are and label the groups. A general report of analysis techniques can be included here, although the information on analysis of auditory features etc is necessary at this point.

p. 12 – Analyses seem to be a multiple analyses for each hypothesis rather than a single hypothesis test. As such the results are difficult to follow and seem rather exploratory rather than confirmatory. I would suggest that the hypotheses and the analyses performed in order to test them be refined and focused in order to provide a clearer pattern of results.

6. PLOS authors have the option to publish the peer review history of their article (what does this mean?). If published, this will include your full peer review and any attached files.

Reviewer #1: No

---

## [Author Response · Author response to Decision Letter 0]

15 Sep 2021

Please see the response to reviewers document enclosed in the submission package.

---

## [Editor Report · Decision Letter 1]

17 Sep 2021

Music and mood regulation during the early-stages of the COVID-19 pandemic

PONE-D-21-11175R1

Dear Dr. Habibi,

We’re pleased to inform you that your manuscript has been judged scientifically suitable for publication and will be formally accepted for publication once it meets all outstanding technical requirements.

Kind regards,

Stefan Koelsch

Academic Editor

PLOS ONE
---

## [Editor Report · Acceptance letter]

27 Sep 2021

PONE-D-21-11175R1 

Music and mood regulation during the early stages of the COVID-19 pandemic 

Dear Dr. Habibi:

I'm pleased to inform you that your manuscript has been deemed suitable for publication in PLOS ONE. Congratulations! Your manuscript is now with our production department. 

Kind regards, 

on behalf of

Prof. Dr. Stefan Koelsch 

Academic Editor

PLOS ONE